# Relationship between Sports and Personal Variables and the Competitive Anxiety of Colombian Elite Athletes of Olympic and Paralympic Sports

**DOI:** 10.3390/ijerph19137791

**Published:** 2022-06-24

**Authors:** Fabián Humberto Marín-González, Iago Portela-Pino, Juan Pedro Fuentes-García, María José Martínez-Patiño

**Affiliations:** 1Faculty of Education and Sport Sciences, University Institution National Sports School, Calle 9 #34-01, Cali 760042, Colombia; fabian.marin@endeporte.edu.co; 2Department of Health Sciences, Isabel I University, 09003 Burgos, Spain; 3Faculty of Sport Science, University of Extremadura, Av. de la Universidad S/N, 10003 Cáceres, Spain; 4Faculty of Sciences of Education and Sport, University of Vigo, 36005 Vigo, Spain; mjpatino@uvigo.es

**Keywords:** high performance, stress, self-confidence, sex, age, type of sport, sport modality, occupation, level

## Abstract

Background: Anxiety is one of the most complex and the most studied constructs in psychology, and it is extremely frequent in high-level sportsmen and women. The main goal was to study the influence of sex, age, type of sport, sport modality, other professional occupation, and competitive level on the competitive anxiety symptoms and self-confidence of elite athletes. Methods: A descriptive cross-sectional study was carried out with Colombian elite athletes who were members of the “Support to the Excellence Coldeportes Athlete” program. The total population studied included 334 Colombian elite athletes: mean age 27.10 ± 6.57 years old with 13.66 ± 6.37 years practicing his/her sports modality. The precompetitive anxiety symptoms of the participants were assessed using the Competitive State Anxiety Inventory—2R (CSAI-2R). Results: Men showed higher levels of self-confidence than women. Younger athletes had a higher cognitive and somatic anxiety. The athletes of individual sports had a higher mean somatic anxiety than those of collective sports. The higher-level athletes had lower values of cognitive and somatic anxiety and higher levels of self-confidence. Finally, the values of anxiety symptoms positively correlated with each other, and negatively correlated with self-confidence. Conclusion: Individualised psychological intervention programs adapted to elite athletes are needed, considering the divergent results found in various variables of scientific interest.

## 1. Introduction

Anxiety is undoubtedly one of the most cited psychological constructs from all the paradigms encompassed in psychology [1], in which somatic and cognitive aspects must be distinguished, including trait and state aspects independent of each other, which influence behaviour in different forms [2]. The multidimensional theory of competitive anxiety [3] proposes three components: somatic anxiety, related with the physiological activation of the organism; cognitive anxiety, associated with the cognitive representations of threat, uncertainty, and worry; and self-confidence [4]. 

Many studies have analysed the possible associations between competitive anxiety symptoms, athlete variables, and the sports context [5,6], with the results from a systematic review and meta-analysis indicating that it may be possible to enhance career longevity and improve the role satisfaction of athletes through focused and acceptable interventions for anxiety symptoms [7]. However, another systematic review indicated insufficient evidence associated to anxiety symptoms and sports performance [8]. In this sense, despite the existence of indicators of cognitive anxiety (e.g., recurrent thoughts or rhyming), coaches and athletes need to understand that there are also indicators of a necessary activation for psychological functioning, with channelling such a process through the psychological training of different skills to enhance the capacities for self-confidence, being very important [9]. Thus, self-confidence is likely to increase as athletes have better levels of acceptance, competence, and cognitive anxiety, and it is likely to reduce their indicators of somatic anxiety [9]. Therefore, cognitive and somatic anxiety are modulated by the athlete’s interpretation of them, which may be beneficial for their performance [10].

The sports psychological profile is able to predict competitive anxiety symptoms, moods, and self-efficacy scores, with self-confidence being the variable that best predicts it [11].

As for the differences that can be established given the sex of athletes, it has been observed that female athletes tend to have higher levels of somatic anxiety as compared to men, as shown by their strong perception of the symptoms that cause their physical condition, such as heart rate, sweating, or the activation of their nervous system. These could condition the athletes’ training, preparation, and competition, and therefore, the somatic anxiety of athletes must be controlled to achieve maximum performance [12].

In relation to competitive anxiety symptoms according to sex, the results from different studies have shown that girls obtain higher scores than boys in competitive anxiety symptoms [9,13,14,15], with higher values on the cognitive anxiety and somatic anxiety scales, and lower values on the self-confidence scale [3,16]. However, although the majority of studies showed higher values of anxiety symptoms in women than men, other studies have not found differences between men and women [17], or when collective sports athletes, such as football [18] or water polo [19], were analysed, with even higher values found in men than in women.

Considering the age variable, different studies have shown that older athletes, compared to younger ones, have higher levels of cognitive anxiety [20,21,22], although it has also been shown that older athletes have better coping strategies and are more likely than their younger counterparts to perceive this type of anxiety as facilitating their performance [20,23]. On the other hand, the results from different studies have shown higher values of self-confidence in older athletes [24,25,26].

On a related note, different studies have shown the influence of a type of sport practiced on competitive anxiety symptoms levels, with higher values of both cognitive and somatic anxiety in individual sports athletes than collective sports [9,15,27,28,29], with lower self-confidence values in individual sports [22].

As for the different levels of anxiety symptoms when comparing elite athletes in adapted and nonadapted sports, the results of a recent study conducted with Olympic and Paralympic athletes during the health crisis caused by COVID-19 did not show significant differences between these populations in terms of feeling nervous/anxious or stressed [30]. The study found anxiety symptoms levels in line with the nonpathological population, showing that the quarantine did not have an impact on the anxiety symptoms response of Olympic and Paralympic athletes [31]. This could be explained by these populations having higher cognitive resources and broader experience in coping with anxiety contexts such as competitions [32]. Another study with Athletes with Disability, who participated at the national level and at the national trials for the Paralympic Games in a variety of sports, showed a similar precompetition anxiety symptoms response as athletes without a disability, although athletes with a disability reported a reduction in self-confidence just prior to the competition [33].

Different studies have analysed the relationship of professional occupation with the prevalence of anxiety and stress, showing that homemakers had 1.2 times higher anxiety and 1.3 times higher stress than working women, and the prevalence of stress and anxiety was also higher in homemakers as compared to working women and students, with the involvement in activities outside the home helping to reduce stress and anxiety [34].

In relation to the influence of the educational level of the athletes on different psychological variables, it seems that resilience and sports courage are positively correlated and important with regard to the educational and sport variables of adolescents [35]. The results of studies carried out with Olympic and Paralympic athletes during the COVID-19 health crisis showed differences depending on the level of education, with athletes with university education having a more positive assessment of the situation than athletes with a lower level of education [31]. Likewise, another study with a similar sample of athletes observed that the group of athletes with professional training was more worried about a reduction in their athletic ability because of the confinement than the group with only a university education [30].

As for anxiety symptoms levels, when comparing athletes with a higher competitive level with others with a lower level, the results of a study showed that high-level athletes obtained lower results in cognitive anxiety, motor anxiety, physiological anxiety, and total anxiety as compared with lower-level athletes before the evaluation, in interpersonal anxiety, and before everyday situations [13]. Similarly, another study showed that high-level athletes had higher levels of self-confidence than lower-level athletes [36].

Given the above, some studies have investigated coping in sports, integrating variables of sex and type of sport [15,27,37], sex and competitive levels [13,36], sex, age, and type of sport [22], and sport type and experience [38]. However, no studies were found that have dealt with the relationship between competitive anxiety symptoms and competitive level together with a large number of variables, such as the present study. Therefore, this study aimed to analyse the competitive anxiety symptoms of Colombian elite athletes of Olympic and Paralympic sports considering sex, age, type of sport, sport modality, other professional occupation, and competitive level.

Based on the review of the literature on the effects of competition on athletes’ anxiety symptoms, it was hypothesised that:

Male athletes have lower anxiety symptoms values and higher self-confidence values than female athletes.


Younger athletes have lower values of anxiety symptoms and higher values of self-confidence as compared to older athletes.Individual sports athletes have higher values of anxiety symptoms and lower self-confidence than collective sports athletes.Paralympic sports athletes have lower values of self-confidence and similar anxiety symptoms values than Olympic sports athletes.Athletes who have no other professional occupations have higher values of somatic anxiety and similar cognitive anxiety and self-confidence values than athletes who have another professional occupation.Athletes with a higher level of education have lower values of anxiety symptoms and higher levels of self-confidence than athletes with lower levels of education.Athletes with a higher competitive level have higher values of self-confidence and lower anxiety symptoms values than athletes with a lower competitive level.Competitive anxiety symptoms values correlate positively. However, they correlate negatively with self-confidence.


## 2. Materials and Methods

The study design is cross-sectional descriptive. The Strengthening the Reporting of Observational Studies in Epidemiology (STROBE) Checklist was utilised, specifically the “STROBE Checklist: cross-sectional studies” (https://www.strobe-statement.org/) (accessed on 16 June 2022). This study was performed online using the Spanish Google Forms Internet platform (https://docs.google.com/forms/) (accessed on 14 August 2020). A correlational design with incidental convenience sampling was used. 

### 2.1. Sample

The inclusion criterion used for the present research was part of the “Support to the Excellence Coldeportes Athlete” program from the Colombian Ministry of Sports. The Coldeportes program’s main objective is to achieve important results by the Colombian high-performance athletes, especially in the Olympic, Paralympic, and Deaflympic Games, and other international competitions. Thus, this program provides technical support to athletes based on sports science. The total population studied included 334 Colombian elite athletes: aged 27.10 ± 6.57 years old, with 3.62 ± 3.12 years within the program, 13.66 ± 6.37 years practicing his/her sports modality, and with a position of 4.05 ± 3.96 achieved in the last international competition: 178 women (26.24 ± 6.25 years) and 156 men (28.10 ± 6.80 years); 177 were 26 years old or older (31.92 ± 5.12 years old) and 157 were 25 years old or younger (21.68 ± 2.59 years old); 287 from individual sports (26.86 ± 6.45 years) and 47 from collective sports (28.74 ± 7.16 years); 284 from Olympic sports (26.16 ± 5.66 years), and 50 from Paralympic sports (32.46 ± 8.59 years). 

The Coldeportes program classifies athletes into the following 7 categories, based on their sports achievements: 1. “Talent” (17 years of age): Gold Medal in the South American/Para-South American Championship or Medallist in the South American Youth Games, or good results in the Youth Para-American Games; 2. “Junior”: Medallist in the Junior World or Pan-American/Parapan American Championships; 3. “Development”: Medallist in South American/Para-South American Sports Games or Silver or Bronze Medallist in Central American and Caribbean Sports Games; 4. “Promotion”: Qualified for Summer or Winter Olympic Games, or Silver or Bronze Medallist in Para/Parapan American Championships, or Medallist in Youth Olympic/Paralympic Games; 5. “Advanced”: Gold Medal in the Pan-American/Parapan American Championship or World Games Medallist; 6. “Elite”: 4th to 8th place in the World Championship, or 6th to 8th place in the World Ranking at the end of the season, or Gold Medal in the Para/Parapan American Games; 7. “Altius”: Medallist in the Summer or Winter Olympic/Paralympic Games, or Medallist in the World Championship, or 1st to 5th place in the world ranking at the end of the season [39].

Table 1 summarizes the descriptive analyses of the independent variables that characterize the sample of athletes under study.

### 2.2. Procedure

The call to participate in the study was made through a link sent by the Directorate of Sports Positioning and Leadership of the Coldeportes program. The questionnaire was sent to all 420 athletes who were part of the program, with 334 (79.52%) answering: 358 from Olympic sports, with 284 (79%) answering, and 62 Paralympic athletes, with 50 (81%) answering. This study was completely voluntary, and no personal data through which the participants could be identified were requested. Athletes were asked to complete the questionnaire 24 h before the start of the competition, the same as in previous studies using the inventory of situations and anxiety questionnaire [13] or the same CSAI-2 used in the present study [11]. The data collection period lasted 21 days (from 2 September to 22 September 2020). Before participating, the procedure to be followed for completing the questionnaire was explained through the aforementioned platform, and the athletes provided their consent before starting it, in accordance with the Declaration of Helsinki [40]. All the procedures were approved by the Ethics Committee of the University Institution “National Sports School” of Cali (Colombia) (approval number: 17.163).

### 2.3. Instruments

The anxiety and self-confidence of the participants were assessed using the Competitive State Anxiety Inventory—2R (CSAI-2R) (Spanish version) [41,42], which is greatly useful for analyses in the sports context [43,44], or even the military [45]. Moreover, the CSAI-2 questionnaire has been shown to be suitable for athletes with disabilities, with the verification of validity and reliability of the CSAI-2 questionnaire carried out for Athletes with Disabilities [46]. The cognitive anxiety, somatic anxiety, and self-confidence values can be extracted from the 17 items of the questionnaire. The questionnaire utilised a 4-point Likert scale, ranging from “not at all” to “very much so” to score the items. The negative feelings about the performance and the results of the performance were assessed with the Cognitive Anxiety subscale. The 5 items of this subscale have an overall score ranging from 5 to 20 points. The perception of anxiety physiological indicators, such as muscle tension, increased heart rate, sweating, and stomach discomfort, was measured with the 7 items of the Somatic Anxiety subscale. The minimum score of this scale is 7, with a maximum of 28. The athletes’ degree of confidence about their success in the competition was measured with the 5 items of the self-confidence subscale, with an overall score between 5 and 20.

### 2.4. Statistical Analysis

A statistical analysis was conducted using the SPSS statistical package (Statistical Package for Social Sciences, version 25 for Windows, IBM Corporation, Armonk, NY, USA) and the level of significance was set at *p* < 0.05.

Next, a reliability analysis was carried out to calculate the internal consistency of the questionnaires. For this, Cronbach’s alpha was used, with a value equal to or greater than 0.70 indicating good consistency [47]. The McDonald omega coefficient was also calculated, which also serves to verify the internal consistency of the variables used in research, and according to some authors, it shows evidence of greater accuracy. In the McDonald omega coefficient, the established range is between 0 and 1, with the highest values providing the most reliable measurements [48]. However, to consider an acceptable value of confidence using the omega coefficient, it must be greater than 0.70 [49].

The analysis of differences in the variables sex (female and male), age (25 years and under or 26 years and over), sport modality (Olympic or Paralympic), type of sport (individual or collective), and, depending on whether the athletes had, in addition to sports, another professional occupation, was carried out using Student’s *t* test for independent samples. For the establishment of comparisons based on the level of education of the athletes (basic education, vocational training, or university education) and the program’s seven categories of athlete classification (Talent to Altius), a one-way ANOVA with Bonferroni’s correction was used. Cohen’s D was used to calculate the effect size. A bivariate correlation analysis was performed with Pearson’s correlation coefficient to test the association between the CSAI-2R variables.

To interpret the result of Cohen’s d, to quantify the magnitude of the effect, small (d = 0.2–0.3), medium (d = 0.5–0.8), and large (d = greater than 0.8) effect values were utilised.

Regarding the interpretation of Pearson’s correlation, it was understood that an r > 0.70 was high and r > 0.90 was very high.

## 3. Results

Table 2 shows the mean values of cognitive anxiety, somatic anxiety, and self-confidence. The internal consistency results suggested an adequate level of internal consistency, with Cronbach’s alpha and McDonald omega coefficient values above 0.80 for all the variables.

The differences found based on the sex variable (female or male) only showed significant differences in the “self-confidence” variable (*p* = 0.005), with males (M = 3.67) having a higher mean than females (M = 3.54) (Table 3).

Considering the age variable, significant differences were found in the variable “cognitive anxiety” (*p* = 0.02), with subjects 25 years of age or younger (M = 2.64) having a higher mean than subjects 26 years of age or older (M = 2.46). On the other hand, in the variable “somatic anxiety” (*p* = 0.001), the 25 years of age or less participants (M = 2.15) obtained a higher mean than those 26 years old or older (M = 1.91) (Table 4).

Considering the sport modality (individual or collective), only significant differences were found in the variable “somatic anxiety” (*p* = 0.009), with the subjects of individual sports (M = 2.05) obtaining a higher average than collective sports (M = 1.80) (Table 5).

Regarding the type of sport (Olympic or Paralympic): significant differences were only found in the “self-confidence” variable (*p* = 0.001), with Paralympic athletes (M = 3.79) having a higher mean than Olympic athletes (M = 3.57) (Table 6).

The analysis of the results depending on whether the athletes had, in addition to sports, another professional occupation, showed that athletes who, apart from being an elite athlete, did not have another professional occupation, had a higher “somatic anxiety” (*p* = 0.045) than those with another professional occupation (Table 7).

When establishing comparisons of the variables based on the different levels of education, significant differences were observed only in the variable “cognitive anxiety” (*p* = 0.042), with the athletes with a basic level of education (M = 2.61) having a higher mean than the subjects with a university education (M = 2.39) (Table 8).

When establishing comparisons of the variables based on the program’s categories, significant differences were found in the variable “cognitive anxiety” between “Junior” (M = 3.05) and “Altius” (M = 2.09) athletes, with higher values in the “Junior” category. There were also significant differences in “somatic anxiety” between “Talent” (M = 2.25) and “Altius” (M = 1.89) athletes, and between “Junior” and “Altius”, in both cases with lower values in “Altius”. There were significant differences in “self-confidence” between the “Talent” (M = 3.47) and the “Elite” (M = 3.78), the “Junior” (M = 3.49) and the “Elite”, and the “Advanced” (M = 3.53) and the “Elite” categories, in all cases with higher values of self-confidence in the “Elite” category (Table 9).

Table 10 shows the correlations between the three CSAI−2R variables. All the correlations were significant, being positive between “Cognitive anxiety” and “Somatic anxiety” and negative between “Self-confidence” and the other two variables: “Cognitive anxiety” and “Somatic anxiety”.

## 4. Discussion

The results showed significant differences in an important number of the dependent variables studied, as well as strong correlations between “Cognitive Anxiety”, “Somatic Anxiety”, and “Self-confidence”.

For the first hypothesis, which referred to differences according to the sex of the athletes, the results obtained partially supported our hypothesis. Although no differences were found in cognitive anxiety and somatic anxiety, the differences in self-confidence found between men and women indicated that men had higher levels of self-confidence. These findings coincide with those obtained when analysing the levels of competitive anxiety symptoms, both trait and state, of young swimmers, where no sex differences were found [17]. In this sense, the results from our study contrast with most of the research up to date, which indicates that anxiety symptoms are greater in female athletes than in male athletes [9,12,13,14,15]. One possible explanation for these results is that the sample in our study was composed entirely of the best elite athletes within a country, as opposed to the aforementioned studies, and top-level athletes may have much more psychological training and experience from years of competition, which allows the equalisation of the levels of anxiety symptoms between men and women. In fact, when male and female Olympic and Paralympic athletes were analysed during the COVID-19 crisis, no differences were found in anxiety symptoms levels [30,31]. Likewise, another study compared male and female soccer professionals, that is, elite players, during the COVID-19 pandemic, and the results indicated that state anxiety and trait anxiety in males were greater than in females, on average [18].

On the other hand, the first hypothesis was accepted, in that men had higher values of self-confidence than women, in the same line as other studies [16]. However, the results were different from those found in other studies in which the CSAI-2R was also used. Thus, no differences in self-confidence according to sex were found when comparing high performance judoka [12], and similarly, using the same questionnaire, with other studies carried out with competitive tennis players [21,50]. In this sense, a meta-analysis conducted to examine the magnitude of sex differences in self-confidence in physical activity showed that in most studies, self-confidence was greater in men than in women when the task was male-oriented or when the situation was competitive, although the age of the subjects and type of confidence measurement employed were also discussed as possible variables contributing to gender differences in self-confidence. [51].

Regarding the second hypothesis, which stated that younger athletes would have lower values of anxiety and higher values of self-confidence than older athletes, the results obtained partially supported this hypothesis. Athletes 25 years of age or younger had higher cognitive and somatic anxieties than subjects 26 years of age or older. In contrast, there were no differences between older or younger athletes in the self-confidence variables. Regarding cognitive and somatic anxiety, the results were contrary to other studies carried out with competitive tennis players, in which players younger than 14 years old showed lower state anxiety and lower somatic anxiety before matches than players older than 14 years old [21], and with younger padel tennis players in the categories U12, U14, and 16 before the beginning of a competition, with the results showing that younger players showed a greater self- confidence [52]. Regarding self-confidence, other studies have found differences between younger and older athletes, in waterpolo players [24], women’s volleyball players [25], and collective sports and individual sports [26]. However, differences have not been found with high level athletes as in the present study. An explanation may be that high level athletes have competed and obtained great results from a young age, becoming the elite within their categories, and this may have equated the results obtained.

As for the third hypothesis, which declares that individual sports athletes would have higher values of anxiety symptoms and lower self-confidence than collective sports athletes, the results obtained partially supported it. Although no differences were found in cognitive anxiety and self-confidence between individual or collective sports, differences were found in the variable somatic anxiety, with the individual sports athletes having a higher average than collective sports. The results on somatic anxiety, but not those on cognitive anxiety, were in line with other studies, such as a meta-analysis on the relationship between the CSAI-2 questionnaire and performance [20], which showed a moderating effect of sport type, such that cognitive and somatic anxiety exerted a greater influence on performance in individual sports. Thus, the results of the present study are consistent with earlier studies, which suggested that individual sport athletes had more intense somatic symptoms, given that the pressure to achieve the desired result is solely theirs [22]. In relation to self-confidence, in which no significant differences were found depending on individual or team sports, these results were not in line with previous studies, in which athletics and basketball athletes were compared, with the latter showing higher levels of self-confidence [53].

Regarding the fourth hypothesis, which stated that Paralympic sports athletes would have lower values of self-confidence and similar anxiety values than Olympic sports athletes, the results obtained partially supported the hypothesis. There were only significant differences in the self-confidence variable, with Paralympic athletes obtaining a higher mean than Olympic athletes. There were no differences between Olympic and Paralympic sports athletes in anxiety symptoms, in line with other studies conducted with Olympic athletes. Significant differences were only found in the self-confidence variable, with Paralympic athletes obtaining a higher mean than Olympic athletes [30,31]. However, contrary to expectations, Paralympic athletes showed higher levels of self-confidence than Olympians, with these results contrary to those from a study with Athletes with Disability who participated at the national level and at the national trials for the Paralympic Games in a variety of sports. This study showed that Paralympic athletes had pre-competition anxiety symptoms response similar to non-Paralympic athletes, although the Paralympic athletes indicated a reduction in self-confidence just before the competition. The authors concluded that, perhaps under competitive stressful situations, these athletes may more often question their confidence to perform at a high level and to achieve their own goals, than do athletes without a disability [33]. In fact, a study carried out with wheelchair netball players and athletes without disabilities showed that the former more frequently stated that one of their main reasons for practicing sports was to ‘‘develop self-confidence’’ [54]. One possible reason for the results from our study is that athletes from elite Paralympic sports, perhaps after having to overcome many obstacles in life, had managed to attain great achievements at the international level within their sports specialty, which would increase their self-confidence.

As for the fifth hypothesis, which specified that athletes who had no other professional occupation would have higher values of somatic anxiety and similar cognitive anxiety and self-confidence values than athletes who had another professional occupation, the results obtained completely supported the hypothesis. Athletes who, apart from being an elite athlete, did not have another professional occupation, had a higher somatic anxiety than those with another professional occupation. These results are consistent with other studies, although not carried out in the field of sports, such as the one that shows that homemakers were most affected and displayed the highest levels of anxiety symptoms and stress, with the prevalence estimates suggesting that employment may help women achieve a better mental health [34]. Thus, having another professional occupation could reduce anxiety symptoms levels in elite sport, although, logically, performance could also be jeopardised by having to spend time and effort on other types of obligations.

Regarding the sixth hypothesis, which stated that athletes with a higher level of education have lower values of anxiety symptoms and higher levels of self-confidence than athletes with a lower level of education, the results obtained partially supported this idea. Athletes with a basic level of education had a higher cognitive anxiety symptoms mean than those with a university education. These results are in line with other studies carried out with Olympic and Paralympic athletes during the crisis caused by COVID-19, which showed that athletes with a university education had a more positive assessment of the situation than athletes with a lower level of education [31], with athletes with professional training having a greater concern about a reduction in their athletic capability because of the confinement than the university-trained athletes [30]. We did not find differences in the self-confidence variable, despite the fact that studies similar to the present one showed that high-school students had a high self-confidence [55]. However, we must specify that in our case, one of the reasons for this lack of differences between these variables may be that the athletes had mainly focused on their sport from a young age, and therefore did not pay much attention to obtaining good academic results. Thus, higher levels of education did not have an influence.

Regarding the seventh hypothesis, on athletes with a higher competitive level having higher values of self-confidence and lower anxiety symptoms values, than athletes with a lower competitive level, the results obtained supported it. The highest category athletes from the “Support to the Excellence Athlete Coldeportes” program (Altius) had lower levels of cognitive anxiety than those from the second lowest category (Junior). On the other hand, the “Altius” athletes also had less somatic anxiety than the athletes at the lowest category of the program (Talent) and the “Junior” athletes. Finally, the athletes from the second highest category (Elite) obtained higher values than the “Talent” athletes, the “Junior” athletes, and even the athletes from the third highest category (Advanced). The greater cognitive and somatic anxiety results found in athletes with a higher competitive level were in line with those from other studies, such as one that analysed the levels of anxiety symptoms between six combat sports of lower, intermediate, and high-level females, and male athletes, in which significant differences were observed between high- versus low-level athletes in total anxiety [13]. On the other hand, the results from our study showed higher levels of self-confidence in higher level athletes, in line with other studies, such as one in which the CSAI-2 questionnaire was also used. This study showed that high-level gymnasts had higher levels of self-confidence than lower-level gymnasts [36], with this finding also supported by other studies with elite athletes, in which high levels of self-confidence were observed in golfers when using the CSAI-2 questionnaire [56], or with cross country skiers and swimmers, also using the CSAI-2R questionnaire [57].

Other studies have also demonstrated the relationship between the anxiety symptoms subscales of the competitive state (cognitive anxiety, somatic anxiety, and self-confidence), and it seems that elite athletes can manage and interpret these anxiety symptoms well [58]. Although somatic anxiety is a conditional response that disappears once competition begins, both it and cognitive anxiety and self-confidence show a significant relationship with performance [59]. Furthermore, it appears that an athlete’s personality may influence his or her cognitive and physiological responses when participating in a competition [60].

Finally, regarding the eighth and last hypothesis, which predicted that competitive anxiety symptoms values would positively correlate with each other, and negatively correlate with self-confidence, the results showed that this hypothesis was fully accepted, as in other studies in which the CSAI-2R questionnaire was also used with competitive athletes [21].

Finally, we consider that our study had some limitations. Given that the data collected were self-reported in its nature, they may be biased due to certain effects such as selective memory, telescope effect, or exaggeration among others. Secondly, we also must consider that the data did not consider longitudinal effects, i.e., the data were not collected at a particular time, and the results may have been affected by an unknown event. Therefore, it would be advisable to carry out these measurements in spaces separated in time to obtain greater reliability.

It will be necessary to expand the hypotheses of the study by broadening the population with a strategic sampling method, after which a multivariate analysis could be conducted to examine and to describe in detail, the simultaneous effect of multiple variables. This would undoubtedly allow us to propose possible lines of action that are much more defined and specific for this population.

## 5. Conclusions

This article analysed competition anxiety symptoms and self-confidence of Colombian elite athletes as a function of sex, age, type of sport, sport modality, other professional occupation, as well as competitive level. The results showed that men had higher levels of self-confidence than women. Athletes 25 years of age or younger had higher cognitive and somatic anxieties than subjects 26 years of age or older. Individual sports athletes had a higher average somatic anxiety than the athletes from collective sports. Paralympic athletes had higher self-confidence values than Olympic athletes. Athletes who, apart from being an elite athlete, did not have another professional occupation, had a higher somatic anxiety score than those with another professional occupation. Athletes with a basic level of education had a higher cognitive anxiety symptoms mean than the subjects with university education. The athletes belonging to the categories of the program with the highest level of sports performance had lower values of cognitive and somatic anxiety, and higher levels of self-confidence. Finally, anxiety symptoms values positively correlated with each other and negatively correlated with self-confidence.

The conclusions of this article allow us to create profiles depending on the characteristics of a person, which in turn will allow us to know the strengths and weaknesses of our athletes in terms of competition anxiety symptoms and self-confidence. Consequently, more focused and personalised interventions can be designed and implemented. 

This knowledge will also allow us to focus on younger athletes to achieve the most optimal profile for competition.

## Figures and Tables

**Table 1 ijerph-19-07791-t001:** Characteristics of the athletes.

Variable	Category		Frequency	Percentage
Sports modality		Olympic	284	85.0
		Paralympic	50	15.0
Type of sport		Collective	47	14.1
		Individual	287	85.9
Mean hours training/week		Less than 10 h	21	6.3
		Between 10 and 15 h	60	18.0
		Between 15 and 20 h	83	24.9
		Between 20 and 25 h	76	22.8
		Between 30 and 35 h	58	17.4
		Between 40 and 45 h	19	5.7
		Between 45 and 50 h	13	3.9
		More than 50 h	4	1.2
Program category		1. Talent	28	8.4
		2. Junior	17	5.1
		3. Development	94	28.1
		4. Promotion	74	22.2
		5. Advanced	60	18.0
		6.Elite	40	12.0
		7. Altius	21	6.3

**Table 2 ijerph-19-07791-t002:** Descriptive statistics and reliability analysis.

Variables	N	Minimum	Maximum	M	SD	α	ω
Cognitive anxiety	334	1.00	4.00	2.54	0.71	0.83	0.83
Somatic anxiety	334	1.00	3.71	2.02	0.63	0.86	0.86
Self-confidence	334	2.20	4.00	3.60	0.42	0.82	0.83

M: Mean, SD: standard deviation, α: Cronbach’s alpha, ω: omega coefficient.

**Table 3 ijerph-19-07791-t003:** Differences between female and male athletes.

Variables	Sex	N	Mean	SD	t	*p*-Value	Effect Size
Cognitive anxiety	Male	156	2.50	0.66	−0.832	0.406	−0.065
	Female	178	2.57	0.75			
Somatic anxiety	Male	156	1.96	0.60	−1.784	0.075	−0.122
	Female	178	2.08	0.65			
Self-confidence	Male	156	3.67	0.38	2.834	0.005 **	0.128
	Female	178	3.54	0.44			

SD: Standard Deviation; *t*: Student’s *t*; * *p*-value < 0.05, ** *p*-value < 0.01.

**Table 4 ijerph-19-07791-t004:** Differences between athletes 26 years old or older and 25 years old or younger.

Variables	Age	N	Mean	SD	t	*p*-Value	Effect Size
Cognitive anxiety	25 or <	157	2.64	0.72	2.253	0.025 *	0.174
	>25	177	2.46	0.69			.
Somatic anxiety	25 or <	157	2. 15	0.62	3.443	0.001 **	0.233
	>25	177	1.91	0.62			
Self-confidence	25 or <	157	3.57	0.39	−0.996	0.320	−0.045
	>25	177	3.62	0.43			

25 or <: subjects 25 years of age or younger, >25: subjects 26 years of age or older; SD: standard deviation; *t*: Student’s *t*; * *p*-value < 0.05, ** *p*-value < 0.01.

**Table 5 ijerph-19-07791-t005:** Differences between individual and collective sports.

Variables	Sports Modality	N	Mean	SD	t	*p*-Value	Effect Size
Cognitive anxiety	Individual	287	2.55	0.71	0.721	0.471	0.081
	collective	47	2.46	0.70			
Somatic anxiety	Individual	287	2.05	0.62	2.619	0.009 **	0.257
	collective	47	1.80	0.57			
Self-confidence	Individual	287	3.60	0.42	−0.339	0.735	−0.022
	collective	47	3.62	0.41			

SD: standard deviation; *t*: Student’s *t*; * *p*-value < 0.05, ** *p*-value < 0.01.

**Table 6 ijerph-19-07791-t006:** Differences between Olympic and Paralympic sports.

Variables	Sport Type	N	Mean	SD	t	*p*-Value	Effect Size
Cognitive anxiety	Olympic	284	2.55	0.70	0.500	0.617	0.054
	Paralympic	50	2.50	0.77			
Somatic anxiety	Olympic	284	2.05	0.62	1.837	0.067	0.176
	Paralympic	50	1.87	0.66			
Self-confidence	Olympic	284	3.57	0.42	−3.481	0.001 **	−0.218
	Paralympic	50	3.79	0.30			

SD: standard deviation; *t*: Student’s *t*; * *p*-value < 0.05, ** *p*-value < 0.01.

**Table 7 ijerph-19-07791-t007:** Differences according to professional occupations.

Variables	Another Professional Occupation	N	Mean	SD	t	*p*-Value	Effect Size
Cognitive anxiety	Yes	70	2.42	0.59	−1.574	0.116	−0.150
	No	264	2.57	0.73			
Somatic anxiety	Yes	70	1.90	0.54	−1.960	0.045 *	−0.165
	No	264	2.06	0.64			
Self-confidence	Yes	70	3.54	0.47	−1.480	0.140	−0.083
	No	264	3.62	0.40			

SD: standard deviation; *t*: Student’s *t*; * *p*-value < 0.05, ** *p*-value < 0.01.

**Table 8 ijerph-19-07791-t008:** Differences by educational levels.

Variables		N	Mean	SD	CI (95%)		Min	Max	F	*p*-Value	Bonferroni
					Lower Limit	Upper Limit					
Cognitive anxiety	BE	213	2.61	0.72	2.51	2.71	1.00	4.00	3.201	0.042	BT-UT = 0.038 *
	VT	26	2.49	0.70	2.21	2.77	1.20	3.80			
	UE	95	2.39	0.68	2.25	2.539	1.00	3.80			
	Total	334	2.54	0.71	2.46	2.61	1.00	4.00			
Somatic anxiety	BE	213	2.08	0.63	1.99	2.16	1.00	3.71	2.462	0.087	No differences
	VT	26	2.00	0.59	1.76	2.24	1.00	3.14			
	UE	95	1.91	0.62	1.78	2.04	1.00	3.57			
	Total	334	2.02	0.63	1.96	2.09	1.00	3.71			
Self-confidence	BE	213	3.58	0.42	3.53	3.64	2.20	4.00	0.736	0.480	No differences
	VT	26	3.67	0.41	3.50	3.8	2.80	4.00			
	UE	95	3.63	0.41	3.54	3.71	2.40	4.00			
	Total	334	3.60	0.42	3.56	3.65	2.20	4.00			

BT: basic education, VT: vocational training, UT: university education; CI: confidence interval; F: variation between sample means; * *p*-value < 0.05, ** *p*-value < 0.01.

**Table 9 ijerph-19-07791-t009:** Differences by athlete categories.

		N	Mean	SD	CI (95%)		Min	Max	F	*p*-Value	Bonferroni
					Lower Limit	Lower Limit					
Cognitive anxiety	1. Talent	28	2.68	0.714	2.40	2.96	1.20	4.00	3.270	0.004 **	2–7 = 0.001
	2. Junior	17	3.05	0.6384	2.72	3.37	1.80	4.00			
	3. Development	94	2.55	0.6826	2.41	2.69	1.20	4.00			
	4. Ascent	74	2.49	0.6655	2.33	2.64	1.00	3.60			
	5. Advanced	60	2.57	0.7786	2.36	2.77	1.00	4.00			
	6. Elite	40	2.47	0.6088	2.28	2.67	1.60	4.00			
	7. Altius	21	2.09	0.7658	1.75	2.44	1.00	3.60			
	Total	334	2.54	0.7090	2.46	2.61	1.00	4.00			
Somatic anxiety	1. Talent	28	2.25	0.6951	1.98	2.52	1.00	3.71	2.291	0.035 *	
	2. Junior	17	2.39	0.6327	2.05	2.70	1.43	3.29			1–7 = 0.033
	3. Development	94	2.05	0.5386	1.94	2.16	1.14	3.29			2–7 = 0.015
	4. Ascent	74	1.92	0.6371	1.77	2.07	1.00	3.57			
	5. Advanced	60	2.02	0.6358	1.85	2.18	1.00	3.57			
	6. Elite	40	1.91	0.6212	1.71	2.11	1.00	3.29			
	7. Altius	21	1.89	0.7310	1.56	2.22	1.00	3.43			
	Total	334	2.02	0.6270	1.96	2.09	1.00	3.71			
Self-confidence	1. Talent	28	3.47	0.4153	3.31	3.63	2.80	4.00	3.235		1–6 = 0.041
	2. Junior	17	3.49	0.5005	3.24	3.75	2.60	4.00			2–6 = 0.033
	3. Development	94	3.56	0.3995	3.48	3.64	2.60	4.00			5–6 = 0.043
	4. Ascent	74	3.65	0.4326	3.55	3.75	2.20	4.00		0.004 **	
	5. Advanced	60	3.53	0.4165	3.42	3.63	2.60	4.00			
	6. Elite	40	3.78	0.3214	3.68	3.89	3.00	4.00			
	7. Altius	21	3.76	0.3774	3.59	3.93	3.00	4.00			
	Total	334	3.60	0.4156	3.56	3.65	2.20	4.00			

CI: confidence interval; F: variation between sample means; * *p*-value < 0.05, ** *p*-value < 0.01.

**Table 10 ijerph-19-07791-t010:** Analysis of correlations between variables.

Variables	Cognitive Anxiety	Somatic Anxiety	Self-Confidence
Cognitive anxiety	1		
Somatic anxiety	0.557 **	1	
Self-confidence	−0.354 **	−0.379 **	1

** The correlation is significant at the 0.01 level (bilateral).

## Data Availability

The data presented in this study are available on request from the corresponding author. The data are not publicly available due to privacy.

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
