# Peer review of "Relationship between Sports and Personal Variables and the Competitive Anxiety of Colombian Elite Athletes of Olympic and Paralympic Sports"

_ijerph, 2022, doi:10.3390/ijerph19137791_

Round 1

Reviewer 1 Report

Dear Authors, I am so interested in an opportunity to review a topic such as “Influence of sports and personal variables on the competitive anxiety of Colombian elite athletes of Olympic and Paralympic sports“. The aim of this study was to analyse the competitive anxiety of Colombian elite athletes of Olympic and Paralympic sports considering gender, age, type of sport, sport modality, other professional occupation, as well as competitive level.

However, I have some concerns.

Major concerns

I would suggest changing the title of the manuscript and refusing to write the word “influence.” L 288: The Authors have not evaluated an “influence.” This single cross-sectional study allows only the correlation between variables to be identified. Additionally, I would suggest that the aim of this study should not be written in the Discussion Unit. Also, I would recommend changing “anxiety” to “anxiety symptoms” or “anxiety symptomatology” throughout the text of the entire manuscript.

I would suggest using the term “sex“ instead of “gender“ throughout the text of the paper. The Authors have not revealed the gender identity of athletes. Although genetic factors typically define a person's sex, gender refers to how they identify on the inside. The word “gender” must be changed to “sex” throughout the manuscript. Unless the Authors can divide athletes according to gender identity (for example, transgender, transsexual, transvestite and etc.).

L 206: It is necessary to specify how the Cohen's D effects sizes and Pearson's correlation coefficients were assessed in the statistical analysis section. References to literature must be updated too.

L 140-142: It seems necessary to clarify the design of this study. According to the data provided by the Authors, this study is cross-sectional in design.

The data in Table 1 appear to be the Results of the study.

Considering that the Authors carried out a single cross-sectional study, I found the manuscript interesting. However, I  have a revision request. I suggest the Authors use STROBE checklist in reporting their cross-sectional study. Please, add the STROBE checklist as a supplementary material and cite it through the main text. You can download the checklist from this link https://www.strobe-statement.org/. Therefore, I have comments concerning the design of the study, the organisation of the study, and the description of the Methodology. These major concerns in the manuscript can be answered in accordance with the STROBE checklist.

Dear Authors, your data analysis is inaccurate. I suggest that you must categorize survey anxiety scores into “low state anxiety” and “high state anxiety” as well as perform bivariate analysis. Additionally, there is the best way to calculate odds ratios and 95% CI for this study design. In order to strengthen your statistical analysis, I would suggest a multivariate analysis too.

L 241, 243, 249….: What do these values such as (.02)“, “(.001)“, “(.009)“ mean?

Tables 5-6: Comparative samples seem to be very different in size as well as I would suggest recalculating these results then using Welch's t-test (for unequal variances).

The Authors write about anxiety throughout the manuscript. However anxiety is a serious mental health disorder. Therefore it is necessary to write very clearly throughout the text of the entire paper about the anxiety symptoms instead of anxiety cases.

The Authors carried out as many as 25 tests to verify the study's hypotheses. Therefore, it is necessary to adjust the p value. It would be optimal to use the Bonferroni type adjustment (https://www.statisticssolutions.com/bonferroni-correction/ ).

L 317: What gender differences do the Authors write about?

The Discussion section does not maintain the Limitation paragraph. I would suggest submitting it.

What do the data in Table 10 give you? Can you discuss the data in the context of anxiety disorder psychopathology?

Conclusions must be rewritten. It seems necessary to generalise the essential results of this study, namely in the Conclusions Unit. Additionally, there is no paragraph of the practical implications, so it must be written.

Minor concerns

The Abstract must be corrected in accordance with the requirements of the IJERPH.

Table 4: I would suggest standardising the rounding of mean values (for example, “2.64” or “2.1474”). It is also necessary to specify the unit of measure of age.

Table 8, Table 9: I would suggest that you refuse to write “minimum” and “maximum”

L 315: “(Pulido et al., 2021)“. It is necessary to unify the citation of references in the text.

The manuscript back matter must be supplemented with information about the Institutional Review Board Statement

Kind Regards

Author Response

Dear Authors, I am so interested in an opportunity to review a topic such as “Influence of sports and personal variables on the competitive anxiety of Colombian elite athletes of Olympic and Paralympic sports“. The aim of this study was to analyse the competitive anxiety of Colombian elite athletes of Olympic and Paralympic sports considering gender, age, type of sport, sport modality, other professional occupation, as well as competitive level.

However, I have some concerns.

Dear reviewer, 

We would like to express our gratitude for the feedback provided, we firmly believe that our manuscript has improved in quality due to the reviewers’ contributions. 

Major concerns

I would suggest changing the title of the manuscript and refusing to write the word “influence.” L 288: The Authors have not evaluated an “influence.” This single cross-sectional study allows only the correlation between variables to be identified. Additionally, I would suggest that the aim of this study should not be written in the Discussion Unit. Also, I would recommend changing “anxiety” to “anxiety symptoms” or “anxiety symptomatology” throughout the text of the entire manuscript.

I would suggest using the term “sex“ instead of “gender“ throughout the text of the paper. The Authors have not revealed the gender identity of athletes. Although genetic factors typically define a person's sexgender refers to how they identify on the inside. The word “gender” must be changed to “sex” throughout the manuscript. Unless the Authors can divide athletes according to gender identity (for example, transgender, transsexual, transvestite and etc.).

We removed the word "influence" from the title.

We removed the objective of the study from the discussion.

We changed "anxiety” to “anxiety symptoms” throughout the text of the entire manuscript, except when the concept "anxiety" was defined generically and when the terms "cognitive anxiety" and "somatic anxiety" were used, in order to respect the terms used by the authors in the investigations in which the Competitive State Anxiety Inventory - 2R (CSAI-2R) was used, as explained in section "2.3. Instruments" referring to perceptions (symptoms or symptomatology).

We have changed the term from gender to sex.

L 206: It is necessary to specify how the Cohen's D effects sizes and Pearson's correlation coefficients were assessed in the statistical analysis section. References to literature must be updated too.

In the statistical analysis section, where we described the statistics used for the inferential and correlational analysis, as well as the reliability of the scale and its factors, the interpretation of Cohen's d and Pearson's correlation has been added to the text.

References 47 and 48 have been updated and replaced by:

Zangaro, G. A., Importance of Reporting Psychometric Properties of Instruments Used in Nursing Research. Western Journal of Nursing Research 2019, 41, (11), 1548-1550.

Zhang, Z. Y.; Yuan, K. H., Robust Coefficients Alpha and Omega and Confidence Intervals With Outlying Observations and Missing Data: Methods and Software. Educational and Psychological Measurement 2016, 76, (3), 387-411.

L 140-142: It seems necessary to clarify the design of this study. According to the data provided by the Authors, this study is cross-sectional in design.

A sentence has been added that defines the study design: "The study design is cross-sectional descriptive".

The data in Table 1 appear to be the Results of the study.

We have included a sentence clarifying that Table 1 summarizes the descriptive analyses of the independent variables that characterize the sample under study.

Considering that the Authors carried out a single cross-sectional study, I found the manuscript interesting. However, I  have a revision request. I suggest the Authors use STROBE checklist in reporting their cross-sectional study. Please, add the STROBE checklist as a supplementary material and cite it through the main text. You can download the checklist from this link https://www.strobe-statement.org/. Therefore, I have comments concerning the design of the study, the organisation of the study, and the description of the Methodology. These major concerns in the manuscript can be answered in accordance with the STROBE checklist.

Thank you very much for the indication. We have downloaded the document "STROBE Checklist: cohort, case-control, and cross-sectional studies (combined)" from https://www.strobe-statement.org/ and we have reviewed all the items. In our manuscript, of the items applicable to our study's design (Descriptive Cross-Sectional), we included all except "(a) Indicate the study's design with a commonly used term in the title or the abstract", which we have already added in the abstract. In addition, we have completed the "STROBE Checklist" and included it as supplementary material. Finally, in the first paragraph of section "2. Materials and methods" we have mentioned that we have used the STROBE Checklist.

Dear Authors, your data analysis is inaccurate. I suggest that you must categorize survey anxiety scores into “low state anxiety” and “high state anxiety” as well as perform bivariate analysis. Additionally, there is the best way to calculate odds ratios and 95% CI for this study design. In order to strengthen your statistical analysis, I would suggest a multivariate analysis too.

We fully understand the reviewer's concern regarding the possibility of categorizing survey anxiety scores into “low state anxiety” and “high state anxiety”, which could provide very interesting information. However, in this specific case, the choice of sample statistics responds to the objectives and hypotheses stated in our research, based on previous studies. Thus, to maintain the methodology of previous studies and facilitate the reproducibility of future studies, anxiety was categorized according to the usual factors suggested by the author of the scale: cognitive anxiety, somatic anxiety, and self-confidence. The resulting variable was used as a scale variable.

The reference of the study followed is:

Andrade Fernández, E. M.; Lois Río, G.; Arce Fernández, C. Propiedades psicométricas de la versión española del Inventario de Ansiedad Competitiva CSAI-2R en deportistas. Psicothema 2007, 19, (1), 150-155.

L 241, 243, 249….: What do these values such as “(.02)“, “(.001)“, “(.009)“ mean?

We have proceeded to incorporate “p=” in the text, as it refers to the p value.

Tables 5-6: Comparative samples seem to be very different in size as well as I would suggest recalculating these results then using Welch's t-test (for unequal variances).

Levene's F test was calculated to test for homogeneity of variances, and as these were equal, we proceeded to use Student's t test.

The Authors write about anxiety throughout the manuscript. However anxiety is a serious mental health disorder. Therefore it is necessary to write very clearly throughout the text of the entire paper about the anxiety symptoms instead of anxiety cases.

As we have commented previously, we changed "anxiety” to “anxiety symptoms” throughout the text of the entire manuscript, except in the cases indicated above.

The Authors carried out as many as 25 tests to verify the study's hypotheses. Therefore, it is necessary to adjust the p value. It would be optimal to use the Bonferroni type adjustment (https://www.statisticssolutions.com/bonferroni-correction/ ).

The Bonferroni test was already used to establish between in which groups there were differences. The results are shown in Table 8 and 9. Due to an omission error in Table 8 we had not written the word "Bonferroni" in the header of the last column. It has now been included.

L 317: What “gender differences“ do the Authors write about?

We have included a sentence in the text providing more details, such as that self-confidence was greater in men than in women when the task was male-oriented or when the situation was competitive, although age of subject and type of confidence measurement employed were also discussed as possible variables contributing to gender differences in self-confidence

The Discussion section does not maintain the Limitation paragraph. I would suggest submitting it..

It has been added.

What do the data in Table 10 give you? Can you discuss the data in the context of anxiety disorder psychopathology?

Table 10 shows the correlations between the three CSAI-2R variables.  As it is logical, all correlations are significant between factors of the same scale, being positive between "Cognitive anxiety" and "Somatic anxiety" and negative between "Self-confidence" and the other two variables: "Cognitive anxiety" and "Somatic anxiety".

The following paragraph has been added:

Other studies have also demonstrated the relationship between the anxiety symptoms subscales of the competitive state (cognitive anxiety, somatic anxiety and self-confidence), and it seems that elite athletes can manage and interpret these anxiety symptoms well (Habibi, Moghaddam, & Soltani, 2017). Although somatic anxiety is a conditional response that disappears once competition begins, both it, and cognitive anxiety and self-confidence, show a significant relationship with performance (Kumar, & Singh, 2918).. Furthermore, it appears that an athlete's personality may influence his or her cognitive and physiological responses when participating in a competition (Balyan, et al. 2016)..

Added in references:

Balyan, K. Y.; Tok, S.; Tatar, A.; Binboga, E.; Balyan, M., The Relationship Among Personality, Cognitive Anxiety, Somatic Anxiety, Physiological Arousal, and Performance in Male Athletes. Journal of Clinical Sport Psychology 2016, 10, (1), 48-58.

Habibi, H.; Moghaddam, A; Soltani, H. Confidence, cognitive and somatic anxiety among elite and non-elite futsal players and its relationship with situational factors. Pedagogics, psychology, medical-biological problems of physical training and sports 2017, 2, 60-64.

Kumar, P.; Singh, A. K. Relationship among somatic anxiety, cognitive anxiety and self confidence with the performance of Jumpers. International Journal of Yogic, Human Movement and Sports Sciences 2018; 3, (2): 619-620.

Conclusions must be rewritten. It seems necessary to generalise the essential results of this study, namely in the Conclusions Unit. Additionally, there is no paragraph of the practical implications, so it must be written.

A paragraph with practical implications has been added.

Minor concerns

The Abstract must be corrected in accordance with the requirements of the IJERPH.

Table 4: I would suggest standardising the rounding of mean values (for example, “2.64” or “2.1474”). It is also necessary to specify the unit of measure of age.

Table 8, Table 9: I would suggest that you refuse to write “minimum” and “maximum”

L 315: “(Pulido et al., 2021)“. It is necessary to unify the citation of references in the text.

The manuscript back matter must be supplemented with information about the Institutional Review Board Statement

The Abstract has been corrected.

The rounding of mean values has been standardised.

Minimum and maximum have been changed to their abbreviation.

The unification of the citation of references in the text has been done.

Reviewer 2 Report

English language need to be improved.

46 and 47 :  coaches and athletes need to understand that there are also indicators of a necessary activation for psychological functioning,

please explain how you can made this statement.

63

141: 

A correlational design with incidental convenience sampling was used.

explain what does it means.

207

A statistical analysis was conducted using the SPSS statistical package 

Statistics methods are reported in different parts of the methods, Better to have a list of the methods used all togheter in order to made it clear which has been used.

You mention level of significane <0.5. Is that for all the statistics used ?

You used both omega coefficient McDonald and Cronbach. Which is the rationale of this ?

Author Response

Dear reviewer, 

We would like to express our gratitude for the feedback provided, we firmly believe that our manuscript has improved in quality due to the reviewers’ contributions. 

English language need to be improved.

The manuscript has also been re-reviewed by the authors and edited by a professional editor.

46 and 47 :  coaches and athletes need to understand that there are also indicators of a necessary activation for psychological functioning,

please explain how you can made this statement.

We refer to the fact that, for example, some studies such as the article cited in lines 46 and 47 [9] (González-Hernández et al., 2020) state that self-confidence is likely to increase as athletes have better levels of acceptance, competence, and cognitive anxiety, and that it is likely to reduce their indicators of somatic anxiety.

For a better explanation, we have added a sentence to the document to support the statement "coaches and athletes need to understand that there are also indicators of a necessary activation for psychological functioning"

63

141: 

A correlational design with incidental convenience sampling was used.

explain what does it means.

This type of sampling is common in the social sciences and is well known to researchers in descriptive studies. It is a non-probabilistic sampling method and it is used when we do not have access to a complete list of the individuals that make up the population under study. Therefore, we do not know the probability that each individual will be selected for the sample, so we proceed to evaluate all the subjects that are within a sample at a specific time.

207

A statistical analysis was conducted using the SPSS statistical package 

Statistics methods are reported in different parts of the methods, Better to have a list of the methods used all togheter in order to made it clear which has been used.

We have tried to include all the methods used in section "2.4. Statistical analysis", also including text clarifying the interpretation of the results of Cohen's d in order to quantify the magnitude of the effect. We have also clarified the interpretation of Pearson's correlation in the same section 2.4.

You mention level of significane <0.5. Is that for all the statistics used ?

Yes, it is the significance level for all statistics used.

You used both omega coefficient McDonald and Cronbach. Which is the rationale of this ?

While it is true that one of the statistics is sufficient, some reviewers suggest calculating both statistics to check for similarity of results.

Round 2

Reviewer 1 Report

The Authors partially answered my questions.

Of course, the design of this study does not allow conclusions to be drawn about the cause-consequence relationship. The Authors underestimated the presence of anxiety symptoms by cut-off point. It is therefore necessary to clarify in the conclusions terms such as “a higher somatic anxiety” (L: 951) to “a higher somatic anxiety score”.

Bivariate analysis is sufficient to answer the hypotheses of the study.

Both object as well as subject of the study seem to be relevant, therefore I recommend accepting this paper. Additionally, I suggest that the Authors deepen their analysis (in terms of multivariate analysis) of this data and make additional recommendations in the future in order to manage the preparation of high-performance Athletes at national level for the Olympics.

Best Regards

Author Response

Dear reviewer,

We would like to express our gratitude for your positive assessment and both the previous and this new feedback provided with the aim of increasing the quality of our manuscript.

Of course, the design of this study does not allow conclusions to be drawn about the cause-consequence relationship. The Authors underestimated the presence of anxiety symptoms by cut-off point. It is therefore necessary to clarify in the conclusions terms such as “a higher somatic anxiety” (L: 951) to “a higher somatic anxiety score”.

The reviewer is indeed right. The wording of that line has been changed to read: "a higher somatic anxiety score".

Bivariate analysis is sufficient to answer the hypotheses of the study.

Both object as well as subject of the study seem to be relevant, therefore I recommend accepting this paper. Additionally, I suggest that the Authors deepen their analysis (in terms of multivariate analysis) of this data and make additional recommendations in the future in order to manage the preparation of high-performance Athletes at national level for the Olympics.

Thank you for your comments. In the limitations of the study, we have added the need to extend the hypotheses of the study by broadening the population with a strategic sampling method, to then carry out a multivariate analysis to examine and detail the simultaneous effect of multiple variables. This would undoubtedly make it possible to propose possible lines of action that are much more defined and specific for this population.